

# Dynamic-Statistic Combined Ensemble Prediction and Impact Factors on China's Summer Precipitation

Xiaojuan Wang[1], Zihan Yang[2*], Shuai Li[3], Qingquan Li[3], Guolin Feng[3, 4*]

College of Electronic and Information Engineering, Changshu Institute of Technology, Suzhou, 215506, China
Department of Atmospheric Sciences, Yunnan University, Kunming, 650500, China
Laboratory for Climate Research, National Climate Center, Beijing, 100081, China
College of Physical Science and Technology, Yangzhou University, Yangzhou, 225009, China

**Abstract** The dynamic-statistic prediction shown excellent performance on monthly and seasonal precipitation prediction in China and has been applied on several dynamical models. In order to further improve the prediction skill of summer precipitation in China, the Unequal-Weighted Ensemble prediction (UWE) based on the dynamic-statistic combined schemes is presented, and its possible impact factors are also analyzed. Results indicate that the UWE has shown promise in improving the prediction skill of summer precipitation in China, on account to the UWE can overcome shortcomings of the structural inadequacy of individual dynamic-statistic prediction, reducing formulation uncertainties, resulting in more stable and accurate predictions. Impact factors analysis indicates that 1) the station-based ensemble prediction with ACC being 0.10-0.11 add PS score being 69.3-70.2, has shown better skills than the grid-based one, as the former produces probability density distribution of precipitation being closer to the observation than the latter. 2) The use of the spatial average removed anomaly correlation coefficient (SACC) may lower the prediction skill and introduce obvious errors on estimating the spatial consistency of prediction anomalies. SACC could be replaced by the revised anomaly correlation coefficient (RACC), which is calculated directly using the precipitation anomalies of each station without subtracting the average precipitation anomaly of all stations. 3) The low dispersal intensity among ensemble samples of UME implies the historical similar error selected by different approach is quite close to each other, making the correction on the model prediction is more reliable. Therefore, the UWE is expected to further improve the accuracy of summer precipitation prediction in China by considering impact factors such as the grid

Zihan Yang
2622261772@qq.com
Guolin Feng
fenggl@cma.gov.cn



or station-based ensemble approach, the method of calculating the ACC, and the
dispersal intensity of ensemble samples in the application and analysis process of
UWE.

**Keywords**: Dynamic-statistic prediction, Unequal weighted ensemble prediction,
Prediction accuracy, Dispersal intensity, Revised anomaly correlation coefficient




**Introduction**

Accurate prediction of summer precipitation across China is paramount for dealing with critical issues such as flood and drought management, economic development, and ensuring food security. However, this task is fraught with challenges due to the intricate interplay among various atmospheric circulation components, including the East Asian summer monsoon (Ding, 1994; Lu, 2005), the Northwest Pacific subtropical high (Tao, 2006), and the East Asia-Pacific teleconnection patterns (Huang, 2004; Huang, 1987). Additionally, external influences, such as the El Niño-Southern Oscillation (ENSO) (Sun et al., 2021) and the snow cover on the Tibetan Plateau (Si and Ding, 2013), further complicate the prediction process. Due to these complexities, increasing the accuracy of summer rainfall prediction in China still faces challenges, the pursuit of more precise summer rainfall predictions in China is an endeavor that warrants the utmost attention from climate scientists (Gong et al., 2016; Wang et al., 2012).

Over the past few decades, there has been a remarkable progression in the foundation of observational data and theoretical understanding, which has significantly enhanced the capabilities of climate dynamical models in predicting seasonal rainfall (Gettelman et al., 2022; Wu et al., 2017). High-resolution climate simulations, such as those with atmospheric resolutions of approximately 50 km and oceanic resolutions of 0.25°, have been successfully implemented by several research institutions (Roberts et al., 2016; Satoh et al., 2014; Wu et al., 2021). These dynamic models have also demonstrated success in long-term prediction of atmospheric circulation patterns and sea surface temperatures in low-latitude regions (Zhu and Shukla, 2013). However, the current performance of seasonal predictions for key climate elements, including rainfall and temperature, particularly in monsoon-influenced areas like East Asia (Gong et al., 2017; Wang et al., 2015), remains somewhat constrained due to inherent limitations in parameterization schemes and the challenges associated with boundary value problems (Wang et al., 2015). This has spurred meteorologists to delve deeper into understanding how to effectively enhance the seasonal prediction skills of climate models to better align with the needs of end-users (Gong et al., 2016). It is well recognized that regional climate characteristics can significantly influence local rainfall patterns. Despite this, dynamic models still struggle to accurately capture these nuances, suggesting that there is potential for improvement in rainfall prediction



through a statistical-dynamic approach (Specq and Batté, 2020). This integrated
methodology could provide a more robust framework for prediction, ultimately
leading to more reliable and actionable climate predictions.

To enhance the precision of rainfall prediction, Chou (1974) initially suggested the
integration of dynamical model data with statistical analogue information. This
approach leverages the prediction errors from historical years with analogous initial
conditions, such as similar circulation anomalies, snow cover, and sea surface
temperatures (SST), to refine dynamic-analogue correction techniques. For instance,
Huang et al. (1993) introduced the evolutionary analogue-based multi-time prediction
method, (Ren and Chou, 2006; Ren and Chou, 2007) employs historical analogue data
to estimate model errors in accordance with the atmospheric analogy principle, (Feng
et al., 2020; Feng et al., 2013) further develops this concept with their correction
method focused on key regional impact factors. Wang and Fan (2009) proposed a
scheme that integrates model forecasts with the observed spatial patterns of historical
"analog years," while Gong et al. (2018) advanced the leading mode-based correction
method. In addition to these advancements, dynamic-statistic correction methods have
been successfully applied to rainfall predictions in regions such as North China (Yang
et al., 2012) and Northeast China (Xiong et al., 2011b). Furthermore, the application
of these dynamic-statistic prediction has been extended to seasonal predictions,
including those for autumn, winter, and spring (Lang and Wang, 2010). At the Beijing
Climate Center, various error selection methods have been operationalized in rainfall
prediction, including the raw field-based similar error selection method, the empirical
orthogonal function-based similar error selection method, the grid-based similar error
selection method, the regional key impact factors-based similar error selection method,
and the abnormal factor-based similar error selection method (Feng et al., 2020).
These innovative approaches underscore the ongoing efforts to harness both
dynamical and statistical insights to achieve more accurate and reliable rainfall
predictions.

Research has consistently demonstrated the benefits of integrating predictions from
multiple climate models. For instance, the Bayesian model averaging approach   (Luo
et al., 2007) and the moving coefficient ensemble approach (Yang et al., 2024) are
two such approaches that have shown promise. The use of a multi-model ensemble



can mitigate the collective local biases that can occur in space, time, and across
different variables when using individual models (Krishnamurti et al., 2016). This
approach not only assigns higher weights to the outputs of more accurate models but
also enhances overall predictive skill and reduces the uncertainty associated with
single-model ensembles (Yan and Tang, 2013). By accounting for comprehensive
uncertainties stemming from both model discrepancies and initial conditions,
multi-model ensembles often outperform single models (Palmer et al., 2004).
Furthermore, the diverse assumptions inherent in different model frameworks can
potentially compensate for our incomplete understanding of atmospheric dynamics
(Yan and Tang, 2013). The multi-model approach has been successfully applied
across a broad spectrum of forecasting needs, including medium-range weather
forecasting (Candille, 2009) and seasonal climate prediction (Vitart, 2006). Given the
aforementioned advantages of dynamic-statistic methods in seasonal predictions, it is
imperative to adopt an ensemble approach that combines the predictions from these
methods. This integration is crucial for further enhancing prediction accuracy and
reliability. By leveraging the collective strengths of various models and techniques,
we can achieve a more robust and nuanced understanding of climate patterns,
ultimately leading to improved prediction capabilities.

In the process of examining the ensemble prediction, it is crucial to take into account
the various factors that can influence its predictive accuracy (Krishnamurti and
Kumar, 2012). The ensemble's output is particularly sensitive to several key elements:
the number of models incorporated, the duration of the dataset utilized for training,
and the distribution of weights for both downscaling and the integration of multiple
models or schemes (Krishnamurti et al., 2016). Both grid-based reanalysis data and
station-based observational data can serve as the foundation for model training or
validation (Ding et al., 2004; Gong et al., 2016; Wang et al., 2015). It is therefore
essential to explore and discuss the differential impact that the use of these two
distinct types of datasets may have on ensemble predictions. Furthermore, the
dispersion of samples across different models or methodologies cannot be overlooked,
as it also affects the ensemble's predictive skill, and deserve certain attention (Houze
et al., 2015).

Based on above statement, the aim of this research is to construct an



Unequal-Weighted Ensemble prediction (UWE) employing a comprehensive array of
dynamic-statistic methods and to explore the potential factors that may influence its
predictive capabilities. Specifically, the study is designed to delve into three primary
areas: (1) Elucidate the process of establishing the UWE through a suite of
dynamic-statistic methods, highlighting the distinctions between grid-based
ensembles and station-based ensembles. (2) Examine the most effective
methodologies for evaluating the spatial congruence between observational data and
the UWE's output. (3) Investigate the connection between the dispersal of samples
across various dynamic-statistic methods and the predictive accuracy of the UWE.
This study will provide a comprehensive analysis of the UWE's development and its
performance, offering valuable insights into the factors that influence its predictive
success.

**1 Data and Method**
**1.1** Data
158       The monthly precipitation data of 1634 stations during 1983–2020 are from the
National Meteorological Information Center of the China Meteorological
Administration. The monthly grid precipitation data during 1983–2020 is derived
from the Combined Rainfall Analysis (CMAP) data of the U.S. Climate Prediction
Center. The model prediction data for summer precipitation in China are hindcast
datasets of the BCC_CPSv3. Monthly climate indices during 1983–2020 including
circulation indices (i.e. AO, AAO), SST indices (i.e. Nino 3.4, Nino 4, Pacific
Decadal Oscillation), snow cover indices (i.e. Tibet snow cover area index, Northeast
China snow cover area index) is available from the Beijing Climate Center website
(http://cmdp.ncc-cma.net/Monitoring/ cn_index_130.php) (Gong et al., 2016).
**1.2** Climate regions division
169       Climate in China influence by various climate systems, such as the Monsoon,
mid-high latitude circulation system and westly jet circulation system etc. (Ding, 1994;
Li et al., 2008; Wu et al., 2017). Since summer rainfall has regional characteristics
and potential impact factors, we divide the whole country into 8 regions (Feng et al.,
2020) in terms of South China (110º~120ºE, 20º~25ºN), East China (110º~123ºE,
25º~35ºN), North China (110º~123ºE, 35º~42.5ºN), Northeast China (110º~135ºE,
42.5º~55ºN), Eastern Northwest China (90º~110ºE, 35º~43ºN), Western Northwest
China (75º~90ºE, 35º~48ºN), Tibet Area (80º~100ºE, 27º~35ºN and Southwest China



(95º~110ºE, 22º~33ºN). Each region is treated separately by the dynamic-statistic
prediction process.

**1.3** The dynamic-statistic predictions
Numerical model is an approximation of the behavior of the actual atmosphere. The
dynamic-statistic prediction is to utilize the information of historical analogues to
estimate model's prediction errors through the statistical method, thereby to
compensate the model deficiencies and reduce the model errors (Huang et al., 1993).
As addressed by Feng et al. (2020), the dynamic-statistic prediction can be explained
by equation (1),
$$\hat{p}(\psi_0) = p(\psi_0) + \tilde{p}(\psi_j) - p(\psi_j), \qquad (1)$$
Where $\hat{p}(\psi_0)$ is the corrected prediction, $p(\psi_0)$ is the original model prediction,
and $p(\psi_j)$ is the model prediction of historical year having the similar initial
conditions as current one, $\tilde{p}(\psi_j)$ is the corresponding historical observation. Eq. (1)
is the integral form of the similarity error correction equation, in which the error term
of the similar historical prediction $\tilde{p}(\psi_j) - p(\psi_j)$ is added to the prediction results of
the numerical model.
$$\hat{p}(\psi_0) \xrightarrow{Estimate} \hat{E}(\psi_0), \qquad (2)$$
The core idea of the dynamic-statistic prediction is developing the scheme how to
select the similar year and estimate historical prediction errors(Feng et al., 2013;
Gong et al., 2016). Eq. (2) transforms improvement in the dynamical model prediction
into the estimation of model error (Feng et al., 2013; Ren and Chou, 2006; Xiong et
al., 2011b).

**1.4** Schemes for the dynamic-statistic prediction
Fig.2 presents the flow chart of the dynamic-statistic prediction method. The key step
is the scheme for selecting the historical similar years, which is the step in the red box.
Different scheme of selecting similar years from the historical dataset corresponds to
different dynamic-statistic prediction scheme. In previous years, a series of the
dynamic-statistic prediction schemes has been developed for selecting similar years
from the historical information, and excellent results have been achieved in predicting
summer precipitation anomalies in China (Feng et al., 2013; Wang and Fan, 2009;



Wang et al., 2015; Xiong et al., 2011b).

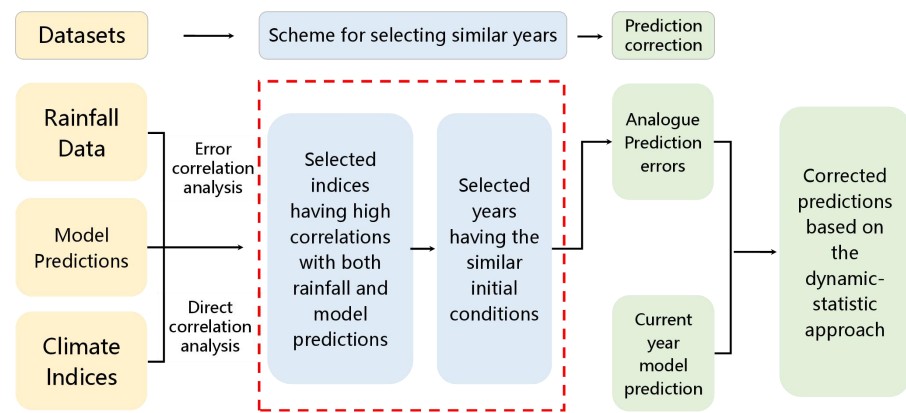


**Fig. 1** The flow chart of the dynamic-statistic prediction method. The key step is the
scheme for selecting the historical similar years, which is the step presented in the red
dash box.

Five kinds of the dynamic-statistic prediction approach representing different
scheme for analogue error selection are introduced as follows,
1) The scheme for original model prediction-based similar error selection (ORM).
With the dynamical model original prediction, select four historical years has the most
similar feature of anomaly distribution as the current year's prediction. Then calculate
the analogue prediction error using these similar years, add to the current prediction
and produce the corrected prediction.
2) The scheme for Empirical Orthogonal Function mode-based similar error
selection (EOF). Calculating the model prediction error filed and produce the
corresponding spatial modes and corresponding principal components using the EOF
method. Similar years is selected based on the Euclidean distance of the principal
components. Historical similar error is calculated using the selected similar years and
added to the current model prediction, which then produce the corrected prediction
(Gong et al., 2018).
3) The scheme for the regional average precipitation-based similar error selection
(REG). Dividing the whole country into 8 regions using according to the introduction
of section 1.2. Selecting the climate indices having high correlations with the regional
average precipitation of each region. With these highly correlated indices,



multi-factors are randomly configured and used to calculate the shortest Euclidean
distance to choose the historical similar years and produce the similar error.
Cross-validation are carried out to correct the model prediction error and obtain the
optimal multi-factor configuration. Based on this final optimal multi-predictor
configuration, the dynamic-statistic prediction can be implemented (Xiong et al.,
2011b).

240        4) The scheme for the grid precipitation-based similar error selection (GRD).
The similar error selection is the same as the REG approach, but the model prediction
error correction is carried out on each grid point within a region.

243        5) The scheme for the abnormal factors based similar error selection (ABN).
Establish factors having significant correlations with the regional precipitation.
Determine the anomaly threshold of each factor and select the key factors reaching
the threshold. Based on the selected abnormal factors, similar years are selected by
the shortest Euclidean Distance of factor set between current year and historical years.
Then the analogue errors can be calculated by using the method of weighted average
integration and be added on the current year's model prediction, which can produce
the corrected prediction (Feng et al., 2020).

251        The selected similar years are not consistent with each other among these five
schemes, the analogue errors usually show similar pattern, but have difference in
detail. Besides the dynamic-statistic prediction, the system error correction are also
presented for comparison.

**1.5** The ensemble for dynamic-statistic prediction

257        Based on the five the dynamic-statistic prediction schemes, the unequal
weighting ensemble prediction (UWE) $E_m$ is calculated as equation (3),
$$E_m = \sum_{k=1}^{n} w_{km} F_{km} \quad (n=5) \ ,$$    (3)
Where $F_{km}$ is the single prediction of each dynamic-statistic scheme and $w_{km}$ is the
weight coefficient of each member. $n$ denotes the total number of dynamic-statistic
scheme, $m$ denotes the current prediction year. $w_k$ can be calculated using equation
263    (4).

$$w_k(i) = \frac{T_k(i)}{\sum_{k=1}^{n} |T_k(i)|} ,$$    (4)



Where $T_k(i)$ is the correlation coefficients between the dynamic-statistic prediction
and observation at each station or grid point $i$. One year out validation is
implemented to define weight coefficients. The anomaly correlation coefficient
(ACC), PS score, and root mean standard error are used for evaluating the prediction
skill for summer precipitation in China. The PS score can be calculated using
equation (5).
$$PS = \frac{f_0 \times N_0 + f_1 \times N_1 + f_2 \times N_2}{N - N_0 + f_0 \times N_0 + f_1 \times N_1 + f_2 \times N_2 + M} \times 100 , \qquad (5)$$

Where $N$ is the total number of stations, is the number of the correctly predicted
stations with abnormal within (-20%, 20%), $f_0$ is weight coefficient of $N_0$; $N_1$ and
$f_1$ are for the stations with abnormal within (-50%, -20%) or (20%, 50%); $N_2$, $f_2$
are for the stations with abnormal within (-100%, -50%) or (50%, 100%); $M$ is the
total number of correctly predicted stations with abnormal below -100% or above
100%. In this study, we set $f_0 = 2$, $f_1 = 2$ and $f_2 = 4$.
Normally, the spatial average removed ACC (SACC) is calculated by formular (6) to
assess the spatial consistency of prediction for summer precipitation in China (Fan et
al., 2012; Xiong et al., 2011b).
$$R = \frac{\sum_{i=1}^{n}(x_i - \bar{x}_s)(y_i - \bar{y}_s)}{\sqrt{\sum_{i=1}^{n}(x_i - \bar{x}_s)^2 \sum_{i=1}^{n}(y_i - \bar{y}_s)^2}} , \qquad (6)$$

Where $n$ is the total number of stations, $x_i$ is the summer precipitation abnormal of
observation at station $i$, while $y_i$ is the summer precipitation abnormal of prediction
at station $i$. $\bar{x}$ and $\bar{y}$ are respectively the average abnormal of observation and
prediction for all the stations. This so-called SACC need to subtract the average
precipitation anomaly of all stations from precipitation anomaly of each station before
calculating the ACC.
In order to confirm if the SACC can properly estimate the spatial consistency of
prediction for summer precipitation, we also calculated the revised anomaly
correlation coefficient (RACC) using formular (7),
$$R^* = \frac{\sum_{i=1}^{n}(x_i^o - \bar{x}_{i,t})(y_i^o - \bar{y}_{i,t})}{\sqrt{\sum_{i=1}^{n}(x_i^o - \bar{x}_{i,t})^2 \sum_{i=1}^{n}(y_i^o - \bar{y}_{i,t})^2}} \qquad (7)$$



Where $n$ is the total number of stations, $x_i^o$ and $y_i^o$ are respectively the summer
precipitation of observation and prediction at station $i$. $\bar{x}_{i,t}$ and $\bar{y}_{i,t}$ is the average of
observation and prediction of summer precipitation for all the years at each station $i$.
The RACC is calculated directly using the precipitation anomalies of each station
without removing the average precipitation anomaly of all stations.

**2 The summer precipitation prediction using the dynamic-statistic scheme**
The RACCs and PSs of the summer precipitation in China produced by the five
dynamic-statistic methods are presented in Table 1. The 10-year average of PS score
of the dynamic-statistic methods varied from 67.4-69.6, which have the better
performance than that of the SYS method (65.8). In figure 2, the temporal correlation
coefficients of the dynamic-statistic methods are higher than the SYS method over
most China with the distribution spatial pattern is similar to each other, but the most
improved areas varied among different method. It is further confirmed with previous
studies that the merger of prediction error estimated via the statistical method and
dynamic model-based original output represents a potential means for improving
prediction skill of summer rainfall in China (Feng et al., 2020).

**Table 1** 10-year average of RACC and PS of the summer precipitation prediction
from 2011 to 2020 for the dynamic-statistic predictions and system error correction.

| Scheme | ORM | EOF | REG |
|--------|-----|-----|-----|
| RACC | 0.10 | 0.03 | 0.01 |
| PS | 69.5 | 69.6 | 67.4 |
| **Scheme** | **GRD** | **ABN** | **SYS** |
| RACC | 0.05 | 0.02 | -0.08 |
| PS | 68.2 | 69.4 | 65.8 |


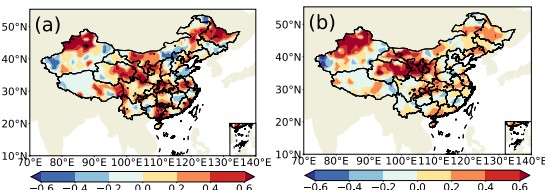






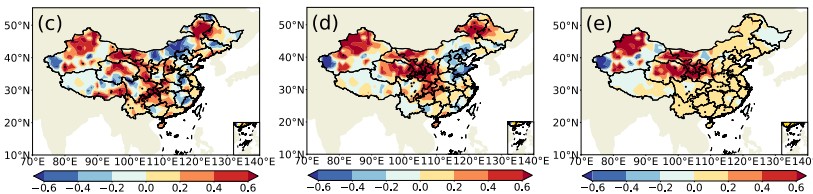

**Fig. 2** The differences of the temporal correlation coefficients for summer precipitation predictions in China from 2011 to 2020. Values indicate differences of the dynamic-statistic method minus the SYS method. (a) ORM, (b) EOF, (c) REG, (d) GRD, and (e) ABN.

Based on the equation of formular (1), four schemes of UWE prediction using the single dynamic-statistic predictions as ensemble members and their corresponding one year out cross validations are presented in Table 2. In order to distinguish the performances UWE prediction against the grid-point observation and station observation, both the grid-based ensemble and station-based ensemble are calculated. Comparing with the single scheme of the dynamic-statistic prediction, the E4 scheme has the best skill among the four ensemble schemes, with RACC being 0.9 and PS score being 70. The grid-based ensemble can somewhat improve the summer precipitation prediction in China, but its effect varied among different schemes. The skills of the station-based ensemble are obviously better than the grid-based one, with RACC being 0.10-0.11 add PS score being 69.3-70.2. As addressed by Yan and Tang (2013) the multi-model ensemble approach (MME) considers the structural inadequacy of individual models and can reduce model formulation uncertainties. The reason why the ensemble of multiple dynamic-statistic predictions can improve the summer precipitation in China is similar to that of MME, which can somewhat overcome the shortcomings of a single prediction and produce the more stable prediction.

**Table 2** 10-year average of RACC and PS score of summer precipitation prediction of the four UWE in China during 2011 ~ 2020.

| Ensemble Scheme | Ensemble member | Grid Ensemble | | Station Ensemble | |
|---|---|---|---|---|---|
| | | RACC | PS | RACC | PS |
| E1 | ORM, GRD | 0.04 | 69.2 | 0.11 | 69.3 |
| E2 | ORM, GRD, EOF | 0.07 | 69.3 | 0.11 | 70.2 |



| | | | | | |
|---|---|---|---|---|---|
| E3 | ORM, GRD, EOF, REG | 0.08 | 69.9 | 0.11 | 70.7 |
| E4 | ORM,GRD,EOF,REG,ABN | 0.09 | 70.0 | 0.10 | 70.1 |


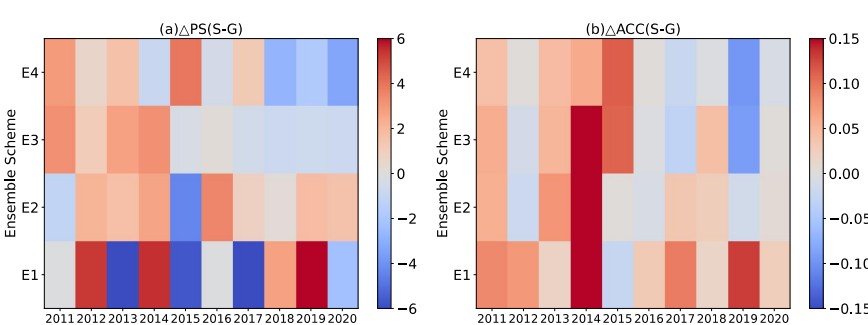


**Fig. 3** Scatter distribution of differences of (a) PS and (b) RACC values for UWE of
summer precipitation in China during 2011 - 2020. Values indicates the differences of
station-based ensemble minus the grid-based ensemble.

In Fig. 4, the TCC of the station-based ensemble for summer precipitation prediction
show positive values in most China, with the high value centers distributed in western
South China, central China, southern North China and western Northeast China etc.
The similar spatial distributions are observed in predictions of the four station-based
ensemble schemes (Fig. 4 a, c, e, g). The TCC differences between the station-based
ensemble and the grid-based ensemble indicate that the former has higher than values
than the later in most areas of China, except for part of Central China and East China
(Fig. 4 b, d, f, h). The spatial distribution of TCC indicates the improvement of the
station-based ensemble is suitable for most stations in China and implies this
approach can make the summer precipitation prediction being closer to the
observation. Bueh et al. (2008) also addressed that the training phase of multi-model
ensemble learns from the recent past performances of models and is used to determine
statistical weights from a least square minimization via a simple multiple regression.
During the training process, more precise objective data can produce better weight
coefficients and lead to more accurate ensemble result, which might be the reason for
the station-based ensemble produce better predictions of summer precipitation in
China than the grid-based one.
Fig.5 indicates that the probability density distribution of station-based ensemble
predictions is closer to the observation especially at the peak part than the gird-based



ensemble and this feature is observed in four ensemble predictions. If the onsite
observation dataset can be used for training, we may have a parameterization scheme
containing precise information for each single station, which may be of help to
produce the prediction being close to the real situation of summer precipitation in
China. Since the gird-based dataset normally is the reproduced observation data,
which may lose certain precise information especially for those extreme values. This
flaw of the grid data may cause it to have poor performance on improving the
prediction accuracy than the station data(Kim et al., 2012; Xiong et al., 2011a; Yang et
al., 2024).



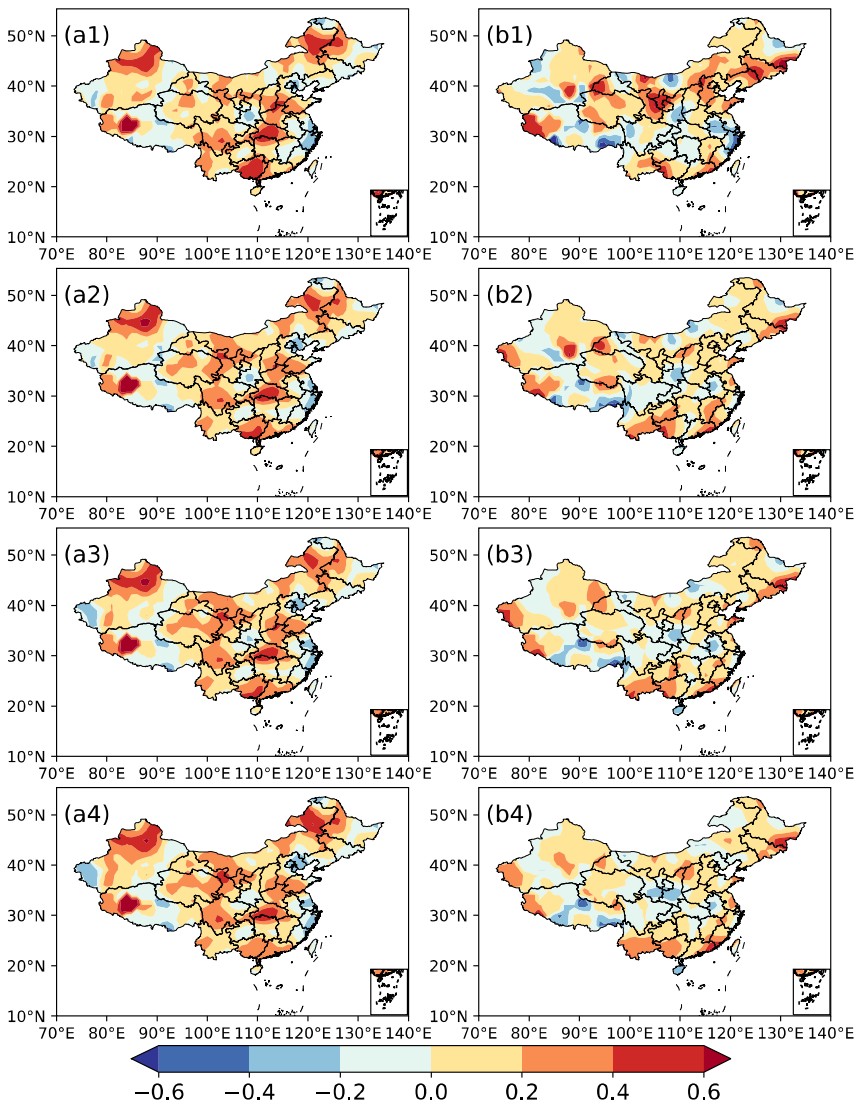

**Fig. 4** Spatial distribution of TCC of station-based UWE for summer precipitation in China during 2011-2020 (a1-a4), TCC differences of station-based ensemble minus the grid-based ensemble (b1-b4). (a1, b1) Ensemble scheme E1; (a2, b3) Ensemble scheme E2; (a3, b3) Ensemble scheme E3; (a4, b4) Ensemble scheme E4.



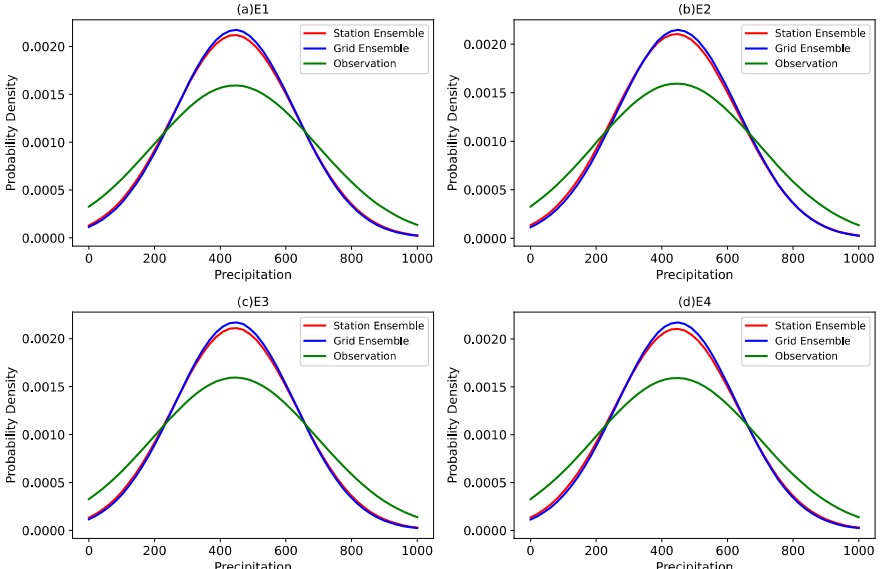

380

**Fig. 5** Probability density distribution of the total precipitation for observation and

UWE. (a) Ensemble Scheme E1, (b) Ensemble Scheme E2, (a) Ensemble Scheme E3,

(a) Ensemble Scheme E4.

**3 Calculating the spatial similarity of ensemble prediction.**

In Fig. 6, the SACCs and RACCs are not consistent with each other, and the former
are more frequently lower than the latter. The 10-year average values of SACC for
each ensemble prediction for summer precipitation in China are also lower than the
RACC (table 1). The SACC is calculated after subtracting the spatial average of
anomaly for all the stations from the original precipitation anomaly. This approach
may cause the new value for each station can't reflect the real situation and lead to a
decrease of RACC between the prediction and observation. In fig.7 the correlation
between the RACC and PS are all higher than those between the SAAC and PS,
which further indicated RACC can better assess the prediction skill of summer
precipitation. It is also noted that the differences between the SACC and RACC are
quite obvious in 2011 and 2015 for ensemble schemes E2, E3, and E4 (Fig. 6 b, c, d).
Comparing with the PS scores, it seems that the RACC for each prediction have more
consistent feature than the SACC. In order to figure out if the RACC has the better
performance than the SACC on indicating the spatial consistency of precipitation
prediction, the observation and prediction of summer precipitation in 2011 and 2015



are respectively presented in Fig. 7. Comparing with the observation (Fig. 7 a5),
predicted precipitation anomalies in summer 2011 show consistent feature in most
China (Fig. 7 a1-a4). The PS scores of four ensemble schemes are respectively 69.5,
68.7, 73.5, 74.3, and RACCs are 0.08, 0.07, 0.10, 0.11, which properly indicate the
prediction skill of these four predictions on the summer precipitation in 2011. It is
also noted that the SACCs of 2011 prediction are respectively 0.01, -0.08, -0.11 and
-0.14, which obviously have flaws in assessing the performance of these four schemes
on predicting the precipitation. This shortcoming of the SACC is also exhibited in the
prediction of summer precipitation anomalies in 2015 (Fig. 7 b1-b5), owing to its
improperly low SACC values being 0.01, -0.07, -0.13, -0.17, respectively.

**Table 3** 10-year average of RACC, SACC of station-based ensemble predictions for
summer precipitation in China during 2011-2020.

|      | E1   | E2   | E3   | E4   |
|------|------|------|------|------|
| RACC | 0.11 | 0.11 | 0.11 | 0.10 |
| SACC | 0.10 | 0.08 | 0.07 | 0.05 |


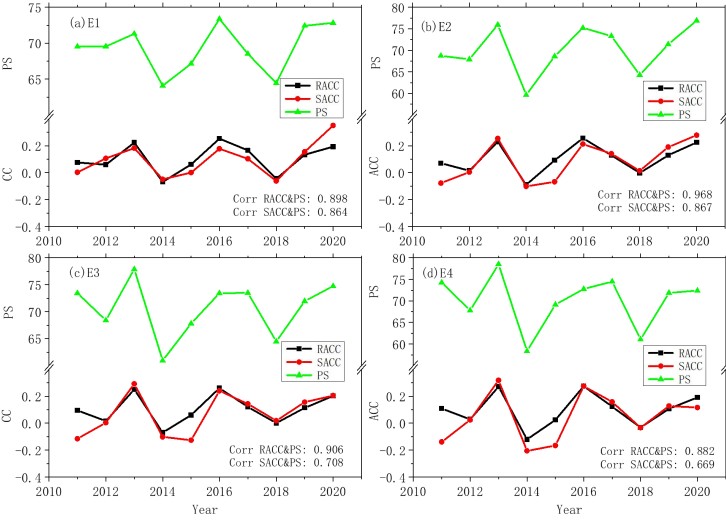


**Fig. 6** Annual **R**ACC, SACC and PS of station-based ensemble predictions for
summer precipitation in China. Prediction of (a) E1, (b) E2, (c) E3 and (d) E4
approach, respectively.

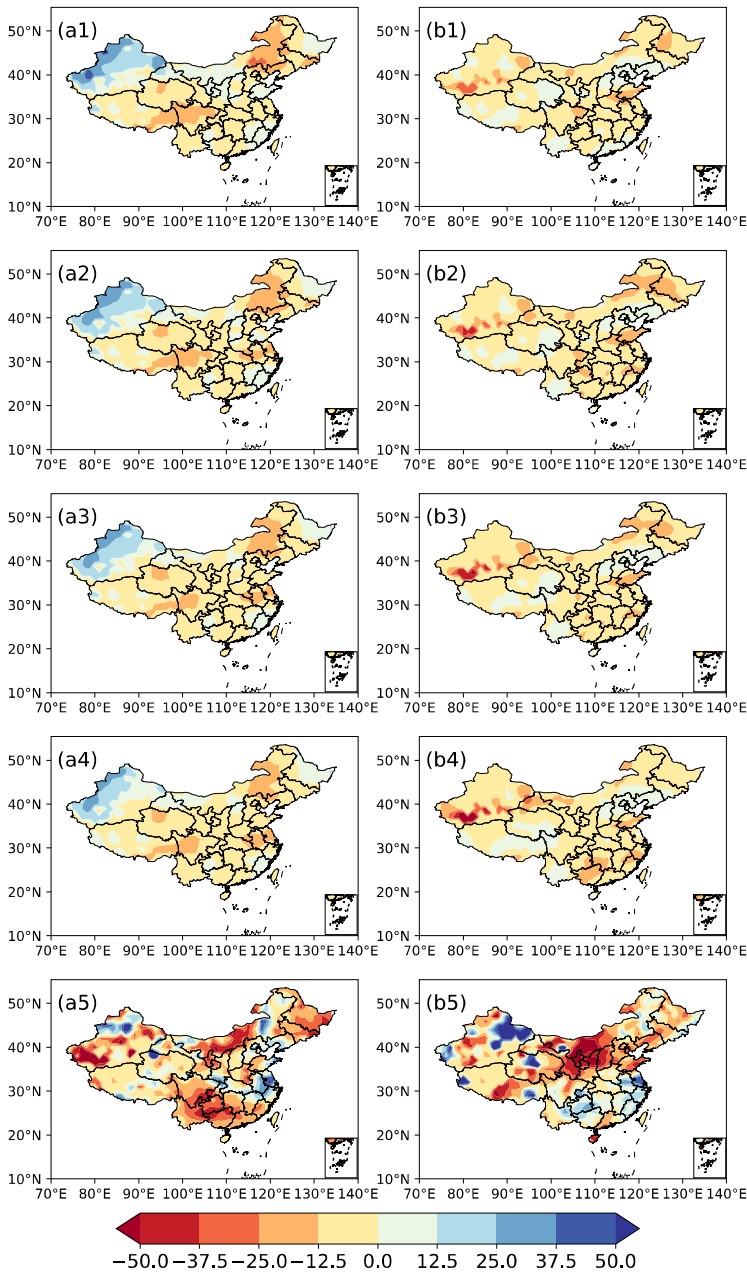


**Fig. 7** The spatial distribution of anomalies (unit: %) of observation and prediction of summer precipitation in 2011 and 2015. (a1-a4) prediction of scheme E1-E4, and (a5) observation for 2011; (b1-b4) prediction of scheme E1-E4, and (b5) observation for 2015.



424

**4. Impact of dispersal intensity on the ensemble prediction.**

The dispersal intensity ($Di$) also called as the coefficient of variation, which is a variable measure the differences among single samples and can be calculated by formal (5). The dispersal intensity is also a relative measure of variability that indicates the size of a standard deviation in relation to its mean. It is a standardized, unitless measure that allows you to compare variability between disparate groups and characteristics.

$$Di = \frac{\sqrt{\sum_{k=1}^{n}(F_{km} - \overline{F}_m)^2 / n}}{\overline{F}_m} \qquad (8)$$

Since the dispersal intensity of each statistic-dynamic prediction has obvious interannual variation, it is necessary to analyze its probable impact on the ensemble prediction of summer prediction in China. Fig. 8 presents the relationship of ACC - dispersal intensity of summer precipitation prediction, in which high ACCs of summer precipitation prediction mostly corresponds to the low dispersal intensity among statistic-dynamic predictions. The variabilities of the signal and noise for the ensemble prediction can be measured as the variance of the ensemble mean and ensemble spread of all the initial conditions (Liu et al., 2019; Zheng et al., 2009),  the sampling error on measuring the signal variance, the more reasonable estimation of the signal variance can be given and used to measure the overall potential predictability of the prediction system (DelSole, 2004; DelSole and Tippett, 2007). The UWE has the similar theory as the ensemble prediction, the low dispersal intensity among ensemble samples implies the historical similar error selected by different approach is quite closet to each other, which makes the correction on the model prediction is more trustable and then produce a more accurate prediction than those cases with high dispersal intensity.








**Fig. 8** The relationship between each UME's ACC and the dispersal intensity of each
summer precipitation prediction during 2011-2020.








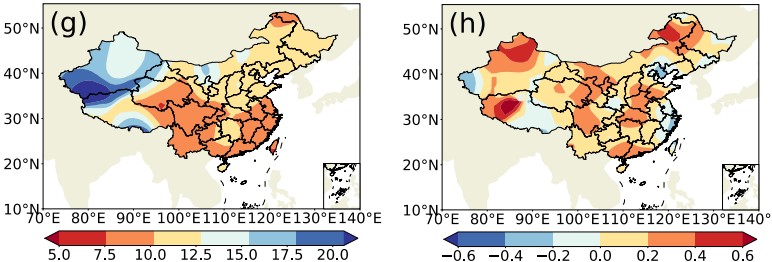

**Fig. 9** The spatial distinction of the 10-year average of dispersal intensity (a, c, e, g) and TCC (b, d, f, h) of UME scheme of E1-E4 during 2011-2020.

In Fig. 9, the 10-year average of dispersal intensity of each UME scheme show the similar pattern as the spatial distribution of TCC of summer prediction produced by UME. Except for part of Northwest China and middle East China, the low dispersal intensity also tends to produce high TCC of statistic-dynamic combined ensemble prediction in most China. The low dispersal intensity among the single prediction corresponds to the major physical process captured by each prediction scheme is similar with each other, which is help of the more reasonable estimation of the signal variance and produce the better precipitation predictions.

**5. Conclusions and discussion**

This study presents the UWE of the dynamic-statistic schemes in order to enhance summer precipitation prediction in China. The analysis also includes an examination of factors that may impact the prediction skill of UWE, such as grid-based and station-based prediction, the calculation of prediction skill, and the influence of sample dispersion on prediction accuracy.

UWE's performance surpasses the model and the dynamic-statistic scheme predictions, potentially due to its ability to overcome individual model or scheme inadequacies, reduce formulation uncertainties, and yield a more stable and accurate predictions. The average RACC and PS values for the station-based ensemble prediction fluctuated between 0.10-0.11 and 69.3-70.2 from 2011 to 2020, indicating significantly higher proficiency compared to the grid-based ensemble prediction. The ensemble prediction based on station data can produce precipitation with a probability density distribution function that is closer to the observed data compared to the



grid-based prediction, making the former more accurate. The use of the SACC needs
to remove the spatial average of the whole stations from the original value, which
may produce inaccurate station values and lead to a lower correlation between
predictions and observations. This makes SACC unsuitable for estimating the spatial
consistency of summer precipitation predictions. The commonly used SACC should
be supplanted by the updated RACC, which is computed by directly utilizing the
precipitation anomalies at each station, without the need to deduct the overall average
precipitation anomaly from all stations.

Moreover, the higher RACCs in summer precipitation prediction are predominantly
associated with lower dispersal intensity among the dynamic-statistic predictions.
This indicates that a more concentrated ensemble, where predictions are closely
aligned, tends to result in more accurate forecasts. Accordingly, the dispersal intensity
of ensemble samples is a crucial factor affecting the prediction accuracy of
dynamic-statistic combined UWE. UWE shares a similar theoretical foundation with
ensemble prediction. Low dispersal intensity among ensemble samples suggests that
the historical similar errors identified by various methods are closely aligned. This
alignment enhances the reliability of corrections applied to model predictions, thereby
yielding more accurate forecasts compared to cases with high dispersal intensities.

**Acknowledgement**,This work is supported by the National Natural Science Foundation of China
Project (Nos.42130610, 42075057, and 42275050) and the National Key Research and Development
Program of China (2022YFE0136000).

**Author contributions.** The conception of this paper is supposed by Xiaojuan Wang and Guolin.
Material preparation, data collection, and analysis were performed by Xiaojuan Wang and Zihan Yang.
The manuscript is written by Xiaojuan Wang and revised by Qingquan Li. All authors commented on
previous versions of the manuscript. All the authors have read and approved the final manuscript.

**Funding**. The National Natural Science Foundation of China Project (Nos. 42130610, 42075057, and
42275050), The National Key Research and Development Program of China (2022YFE0136000).
**Data Availability Statement**. The datasets generated during and/or analyzed during the current study
are available from the corresponding author on reasonable request.
**Ethics declarations.** The authors declare no conflict of interest. Authors have no additional
information that might be relevant for the editors, reviewers and readers. The funding sponsors have no



participation in the execution of the experiment, the decision to publish the results, nor the writing of
the manuscript.

**Conflict of interest:** The authors have no relevant financial or non-financial interests to disclose.

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
