# Peer review of "Dynamic-Statistic Combined Ensemble Prediction and"

_EGUsphere, 2024_

## Referee Comment (RC1)

The Unequal-Weighted Ensemble prediction (UWE) based on the dynamic-statistic combined schemes is presented, and its possible impact factors are also analyzed in the paper. It is indicated that it's a meaningful method for improving summer precipitation in China. Uncertainty analysis also provides key information on common issues needed to be further discussed in the climate prediction area and provides quite meaningful results. I would like to suggest a moderate revision before this manuscript can be accepted for publication. My comments are listed as follows,

1.In the abstract, it is suggested to revise the sentence "the Unequal-Weighted Ensemble prediction (UWE) based on the dynamic-statistic combined schemes is presented," as "the Unequal-Weighted Ensemble prediction (UWE) using outputs of the dynamic-statistic prediction is presented,", which makes it more concise and understandable. This change could be considered in the whole manuscript.

2.The title of part 1.5 "The ensemble for dynamic-statistic prediction" could be revised as "The Dynamic-Statistic Combined Ensemble Prediction". Then the definition of "Dynamic-Statistic Combined Ensemble Prediction" needs to be briefly explained in this part.

3. In the figure 1, the explanation of climate indices needs to be addressed in the title of figure 1.

4. The meaning of $w_k$ needs to be addressed.

5. The title of figure 3 "Scatter distribution of differences of (a) PS and (b) RACC values for UWE of summer precipitation in China during 2011 - 2020.", suggest revising as "Scatter distribution of differences of (a) PS and (b) RACC for summer precipitation prediction during 2011 – 2020 between station-based and grid-based UWE."

6. The dispersal intensity ($Di$) is proposed authors or referenced from other studies need to addressed in the manuscript.

7.In figure 8, why there are four same color circles for each year. Further description needs to be added in the figure title.

8. discussion why the UWE prediction can further improve summer rainfall prediction in China needs to be addressed in the last part of the manuscript.

---

## Referee Comment (RC2)

Comments to the Authors

Review of "Dynamic-Statistic Combined Ensemble Prediction and Impact Factors on China's Summer Precipitation" by Wang et al., submitted to *Nonlinear Process in Geophysics*.

The study addresses how to improve the prediction skills of summer precipitation in China. The authors introduced an approach, the Unequal-Weighted Ensemble prediction (UWE) which was based on the dynamic-statistic combined schemes, and found that the UWE scheme was promising in improving the prediction skills. In addition, the authors also analyzed some possible impact factors, such as the choice of station-based or grid-based datasets, the method of calculating the ACC and the dispersal intensity.

The paper is interesting and well written, and attempt to explore how to improve the prediction skills of summer precipitation in China. However, some revisions are needed for this work. It can be recommended to publish after the following issues have been well responded. Detailed comments are below.

1. The authors reviewed some papers on obtaining the model error information from the historical datasets in the introduction. In fact, the initial condition errors are also important as the model errors, which have large impacts on the prediction skills. Likewise, some researchers also employed the historical datasets to determine the initial condition errors, such as the nonlinear local Lyapunov exponent (NLLE) method or the local dynamical analogs (LAD) method. I think if the authors could add some review on initial condition errors from NLLE or LAD, it will enrich the introduction. Some papers on NLLE or LAD are as follows,

   Li J, Ding R. Temporal–spatial distribution of atmospheric predictability limit by local dynamical analogs[J]. Monthly Weather Review, 2011, 139(10): 3265-3283.

   Ding R, Li J, Seo K H. Predictability of the Madden–Julian oscillation estimated using observational data[J]. Monthly Weather Review, 2010, 138(3): 1004-1013.

   Li X, Ding R, Li J. Quantitative study of the relative effects of initial condition and model uncertainties on local predictability in a nonlinear dynamical system[J]. Chaos, Solitons & Fractals, 2020, 139: 110094.

   Li X, Ding R, Li J. Quantitative comparison of predictabilities of warm and cold events using the backward nonlinear local Lyapunov exponent method[J]. Advances in Atmospheric Sciences, 2020, 37: 951-958.

2. The authors presented the unequal-weighted Ensemble prediction (UWE) scheme based on five dynamic-statistic prediction approaches. The unequal-weighted coefficients have significant effects on its performances in prediction skills. Therefore, what are the criteria for determining the unequal-weighted coefficients? More explanations are appreciated.

3. In comparison of the prediction skills between the grid-point observation and station observation, the authors have described the difference between the two observations based on the information from Table 2. However, the authors seemed to lack the descriptions of the fig.3.

4. In fig.5, the probability density distribution of the total precipitation is the normal distribution. Generally, the precipitation shows the skewed distribution. What is the reason, could the authors give some reasons?

5. The authors investigated the relationship between the dispersal intensity and the ACC, and found that high ACCs corresponds to low dispersal intensity. That is, they have the negative correlation. From fig.8, most regions over China, such as the northwest region and coastal areas show the negative correlations. However, the negative correlations are not that evident in the middle regions of China. What is the reason, some clarifications are needed.

Minor comments
1. Lines 86-87, the reference format is not correct. Right format is Feng et al. (2013, 2020).

2. Lines 170 and 348, before 'etc', there should be a comma.

3. Line 202, Fig.2 should be replaced with Fig.1.

4. For equation (6), what the $x_s$, $y_s$, $\bar{x}_s$ and $\bar{y}_s$ represent? They should be clarified.

5. Line 302 and other part throughout the manuscript, I don't find what the SYS method represents. Is it the abbreviation? If so, please add its full name at the Line 302. If not, please introduce the SYS briefly in the corresponding section.

6. Line 428, it is formula (8), not 'formal (5)'.

7. Line 446, it is quite close, not 'it is quite closet'.

---

## Author Comment (AC1)

Dear Editors and Reviewers:

Thank you for your letter and for the reviewers' comments concerning our manuscript entitled "Dynamic-Statistic Combined Ensemble Prediction and Impact Factors on China's Summer Precipitation". Those comments are all valuable and very helpful for revising and improving our paper, as well as the important guiding significance to our research. We have studied comments carefully and have made corrections which we hope to meet with approval. Our point-by-point response to comments is listed below. All changes are highlighted in blue in the manuscript.

Looking forward to hearing from you.
Thank you and best regards.

Yours sincerely,
Zihan Yang

**Reviewer 1**

**1.In the abstract, it is suggested to revise the sentence "the Unequal-Weighted Ensemble prediction (UWE) based on the dynamic-statistic combined schemes is presented," as "the Unequal-Weighted Ensemble prediction (UWE) using outputs of the dynamic-statistic prediction is presented,", which makes it more concise and understandable. This change could be considered in the whole manuscript.**

**Reply:** Thanks! We have made modifications throughout the entire manuscript.

**The expression in line 12 has been modified as follows:**

In order to further improve the prediction skill of summer precipitation in China, the Unequal-Weighted Ensemble prediction (UWE) using outputs of the dynamic-statistic prediction is presented, and its possible impact factors are also analyzed.

Relevant changes have been modified in the whole manuscript.

**2. The title of part 1.5 "The ensemble for dynamic-statistic prediction" could be revised as "The Dynamic-Statistic Combined Ensemble Prediction". Then the definition of "Dynamic-Statistic Combined Ensemble Prediction" needs to be briefly explained in this part.**

**Reply**: Thanks! We have amended the article in the appropriate section and added a brief description of the relevant definitions.

**The expression in line 259 has been modified as follows:**

**1.5** The Dynamic-Statistic Combined Ensemble Prediction

In order to further improve the effectiveness of summer precipitation prediction by various dynamic-statistic schemes, this study conducted the Dynamic-Statistic Combined Ensemble Prediction called Unequal-Weighted Ensemble prediction (UWE).

Based on the five the dynamic-statistic prediction schemes, the unequal weighting ensemble prediction (UWE) $E_m$ is calculated as equation (3),

$$E_m = \sum_{k=1}^{n} w_{km} F_{km} \quad (n=5) \quad ,$$ (3)

Where $F_{km}$ is the single prediction of each dynamic-statistic scheme and $w_{km}$ is the weight coefficient of each member. $n$ denotes the total number of dynamic-statistic scheme, $m$ denotes the current prediction year. $w_{km}$ can be calculated using equation (4). Using a method similar to the cross check, the TCC was calculated by removing the precipitation predictions of the screened members along with the precipitation actuals for the $m$ th ($m \in [1,10]$) year of data.

$$w_{km} = \frac{T_{km}}{\sum_{k=1}^{n} |T_{km}|}$$ (4)

The weights $w_{km}$ were calculated for each member at each grid-point in the year $m$, where $T_{km}$ is the TCC value calculated for the $k$nd member at that station or grid point after excluding the precipitation data in year $m$, and $w_{km}(k=1,2,...,n)$ is the

weight of the *k*rd member at that grid-point in year  $m$ .

**3.  In the figure 1, the explanation of climate indices needs to be addressed in the title of figure 1.**

**Reply**: Thanks! We have modified the title of Fig.1 to include a description of the climate indices.

**The expression in line 209 has been modified as follows:**

[Figure]

**Fig. 1** The flow chart of the dynamic-statistic prediction method. The key step is the scheme for selecting the historical similar years, which is the step presented in the red dash box. The climate indices refer to the 130 monthly climate indices in terms of the SST indices, Circulation indices etc. during 1983－2020 in section 1.1

**4.  The meaning of  $w_k$   needs to be addressed.**

**Reply**: Thanks! We have modified the description of the relevant formula and add the explanation of  $w_k$ .

**The expression in line 264 has been modified as follows:**

Based on the five the dynamic-statistic prediction schemes, the unequal weighting ensemble prediction (UWE) $E_m$   is calculated as equation (3),

$$E_m = \sum_{k=1}^{n} w_{km} F_{km} \quad (n=5)$$ ,     (3)

Where  $F_{km}$  is the single prediction of each dynamic-statistic scheme and  $w_{km}$   is the weight coefficient of each member, which indicates the contribution of single

dynamic-statistic prediction in the ensemble process. $n$ denotes the total number of dynamic-statistic scheme, $m$ denotes the current prediction year. $w_{km}$ can be calculated using equation (4), which implies higher skill prediction scheme be given larger weight coefficient. Using a method like the cross validation, the TCC was calculated by removing the precipitation predictions of the screened members along with the precipitation actuals for the $m$th ($m \in [1,10]$) year of data.

$$w_{km} = \frac{T_{km}}{\sum_{k=1}^{n} |T_{km}|}$$

(4)

The weights $w_{km}$ were calculated for each member at each grid-point in the year $m$, where $T_{km}$ is the TCC value calculated for the $k$nd member at that station or grid point after excluding the precipitation data in year $m$, and $w_{km}(k=1,2,...,n)$ is the weight of the $k$rd member at that grid-point in year $m$.

**5. The title of figure 3 "Scatter distribution of differences of (a) PS and (b) RACC values for UWE of summer precipitation in China during 2011 - 2020.", suggest revising as "Scatter distribution of differences of (a) PS and (b) RACC for summer precipitation prediction during 2011 – 2020 between station-based and grid-based UWE".**

**Reply**: Thanks! We have changed the title of this figure.

**The expression in line 349 has been modified as follows:**

[Figure]

**Fig. 3** Scatter distribution of differences of (a) PS and (b) RACC for summer precipitation prediction during 2011-2020 between station-based and grid-based UWE. Values indicate the differences of station-based ensemble minus the grid-based

**6. The dispersal intensity (Di) is proposed authors or referenced from other studies need to be addressed in the manuscript.**

**Reply**: Thanks! We have added references where relevant to the article.

**The expression in line 452 has been modified as follows:**

The dispersal intensity (Di) also called as the coefficient of variation, which is a variable measure the differences among single samples and can be calculated by formal (5). The dispersal intensity is also a relative measure of variability that indicates the size of a standard deviation in relation to its mean. It is a standardized, unitless measure that allows us to compare variability between disparate groups and characteristics (Tyralis and Papacharalampous 2024).

**7. In figure 8, why there are four same color circles for each year. A further description needs to be added in the figure title.**

**Reply**: Thanks! We've added a more detailed description in the figure title.

**The expression in line 474 has been modified as follows:**

[Figure]

**Fig. 8** The relationship between each UME's ACC and the dispersal intensity of each summer precipitation prediction during 2011-2020. The four dots of each color indicate the four schemes (E1-E4) applied in ear year's dynamic-statistic prediction.

**8. Discussion on why the UWE prediction can further improve summer**

**rainfall prediction in China needs to be addressed in the last part of the manuscript.**

**Reply**: Thanks! We've added a discussion at the end of the manuscript.

**The expression in line 511 has been modified as follows:**

UWE's performance surpasses the model and the dynamic-statistic scheme predictions, potentially due to its ability to overcome individual model or scheme inadequacies, reduce formulation uncertainties, and yield a more stable and accurate predictions. The average RACC and PS values for the five dynamic-statistic schemes that were ensemble members are 0.02-0.10 and 67.4-69.6. In contrast, the grid-based ensemble prediction of UWE becomes 0.04-0.09 and 69.2-70.9, which is an improvement compared to the dynamic-statistic schemes. Station-based ensemble prediction shows superior performance for this well compared to grid-based ensemble prediction and dynamic-statistic methods, achieving average RACC values of 0.10-0.11 and PS values of 69.3-70.7.

**Reviewer Anonymous Referee**

**1. The authors reviewed some papers on obtaining the model error information from the historical datasets in the introduction. In fact, the initial condition errors are also important as the model errors, which have large impacts on the prediction skills. Likewise, some researchers also employed the historical datasets to determine the initial condition errors, such as the nonlinear local Lyapunov exponent (NLLE) method or the local dynamical analogs (LAD) method. I think if the authors could add some review on initial condition errors from NLLE or LAD, it will enrich the introduction.**

**Reply**: Thanks! We have revised the introduction and cited relevant literature.

**The introduction has been modified as follows:**

Accurate prediction of summer precipitation across China is paramount for dealing with critical issues such as flood and drought management, economic development, and ensuring food security. However, this task is fraught with challenges due to the intricate interplay among various atmospheric circulation components, including the East Asian summer monsoon (Ding 1994, Lu 2005), the

Northwest Pacific subtropical high (Tao 2006), and the East Asia-Pacific teleconnection patterns (Huang 1987, Huang 2004). Additionally, external influences, such as the El Niño-Southern Oscillation (ENSO) (Sun, Yang et al. 2021) and the snow cover on the Tibetan Plateau (Si and Ding 2013), further complicate the prediction process. Some studies have also shown that improving the Real-time Multivariate Madden-Julian Oscillation (RMM) index or introducing better intraseasonal signal extraction methods may allow for higher predictability limits in real-time forecasting (Ding and Seo 2010). Due to these complexities, increasing the accuracy of summer rainfall prediction in China still faces challenges, the pursuit of more precise summer rainfall predictions in China is an endeavor that warrants the utmost attention from climate scientists (Wang, Schepen et al. 2012, Gong, Hutin et al. 2016).

Over the past few decades, there has been a remarkable progression in the foundation of observational data and theoretical understanding, which has significantly enhanced the capabilities of climate dynamical models in predicting seasonal rainfall (Wu, Vitart et al. 2017, Gettelman, Geer et al. 2022). High-resolution climate simulations, such as those with atmospheric resolutions of approximately 50 km and oceanic resolutions of 0.25°, have been successfully implemented by several research institutions (Satoh, Tomita et al. 2014, Roberts, Hewitt et al. 2016, Wu, Yu et al. 2021). These dynamic models have also demonstrated success in long-term prediction of atmospheric circulation patterns and sea surface temperatures in low-latitude regions (Zhu and Shukla 2013). However, the current performance of seasonal predictions for key climate elements, including rainfall and temperature, particularly in monsoon-influenced areas like East Asia (Wang, Fan et al. 2015, Gong, Dogar et al. 2017), remains somewhat constrained due to inherent limitations in parameterization schemes and the challenges associated with boundary value problems (Wang, Fan et al. 2015). This has spurred meteorologists to delve deeper into understanding how to effectively enhance the seasonal prediction skills of climate models to better align with the needs of end-users (Gong, Hutin et al. 2016). It is well recognized that regional climate characteristics can significantly influence local rainfall patterns, atmospheric predictability varies significantly between regions, altitudes and seasons (Li and Ding 2011). Despite this, dynamic models still struggle to accurately capture these nuances, suggesting that there is potential for improvement in rainfall prediction through a statistical-dynamic approach (Specq and Batté 2020).

This integrated methodology could provide a more robust framework for prediction, ultimately leading to more reliable and actionable climate predictions. The relative impact of initial condition and model uncertainties on local predictability also varies with the system state. Therefore, strategically reducing uncertainties in sensitive regions can effectively improve forecasting skills (Li, Ding et al. 2020). Apart from that, warm events are easier to predict than cold events (Li, Ding et al. 2020).

**2. The authors presented the unequal-weighted Ensemble prediction (UWE) scheme based on five dynamic-statistic prediction approaches. The unequal-weighted coefficients have significant effects on its performance in prediction skills. Therefore, what are the criteria for determining the unequal-weighted coefficients? More explanations are appreciated.**

**Reply**: Thanks! We've added a more detailed description, including the meaning of $w_k$, in the relevant section of the manuscript.

**The expression in session II**

"Based on the five the dynamic-statistic prediction schemes, the unequal weighting ensemble prediction (UWE) $E_m$ is calculated as equation (3),

$$E_m = \sum_{k=1}^{n} w_{km} F_{km} \quad (n = 5),$$ (3)

Where $F_{km}$ is the single prediction of each dynamic-statistic scheme and $w_{km}$ is the weight coefficient of each member. $n$ denotes the total number of dynamic-statistic scheme, $m$ denotes the current prediction year. $w_k$ can be calculated using equation (4).

$$w_k(i) = \frac{T_k(i)}{\sum_{k=1}^{n} |T_k(i)|},$$ (4)

Where $T_k(i)$ is the correlation coefficients between the dynamic-statistic prediction and observation at each station or grid point $i$. One year out validation is implemented to define weight coefficients."

**has been modified as follows:**

Based on the five the dynamic-statistic prediction schemes, the unequal weighting ensemble prediction (UWE) $E_m$ is calculated as equation (3),

$$E_m = \sum_{k=1}^{n} w_{km} F_{km} \qquad (n=5) \quad ,$$  (3)

Where $F_{km}$ is the single prediction of each dynamic-statistic scheme and $w_{km}$ is the weight coefficient of each member, which indicates the contribution of single dynamic-statistic prediction in the ensemble process. $n$ denotes the total number of dynamic-statistic scheme, $m$ denotes the current prediction year. $w_{km}$ can be calculated using equation (4), which implies higher skill prediction scheme be given larger weight coefficient. Using a method similar to the cross check, the TCC was calculated by removing the precipitation predictions of the screened members along with the precipitation actuals for the $m$ th ($m \in [1,10]$) year of data.

$$w_{km} = \frac{T_{km}}{\sum_{k=1}^{n} | T_{km} |}$$  (4)

The weights $w_{km}$ were calculated for each member at each grid-point in the year $m$, where $T_{km}$ is the TCC value calculated for the $k$nd member at that station or grid point after excluding the precipitation data in year $m$, and $w_{km}(k=1,2,...,n)$ is the weight of the $k$rd member at that grid-point in year $m$.

**3. In comparison of the prediction skills between the grid-point observation and station observation, the authors have described the difference between the two observations based on the information from Table 2. However, the authors seemed to lack the descriptions of the fig.3.**

**Reply**: Thanks! We have added a description of Figure 3 at the appropriate place in the article.

**The expression in line 355 has been added as follows:**

Figure 3 shows that there is no significant difference between the PS scores of the station-based and grid-based UWE for the E1 scheme. However, for the E2 scheme, the station-based UWE clearly outperforms the grid-based UWE. Additionally, the station-based UWE of E3 and E4 schemes also has relatively higher PS scores. RACC values, of the station-based prediction for all four schemes are

generally higher than those for the grid-based UWE. In summary, compared to the grid-based UWE, the station-based one have obvious higher PS scores and RACC values, indicating better prediction performance.

**4. In fig.5, the probability density distribution of the total precipitation is the normal distribution. Generally, the precipitation shows a skewed distribution. What is the reason, could the authors give some reasons?**

**Reply**: Thanks! We've added a brief description in the relevant section of the article.

**The expression in line 390 has been added as follows:**

The station-based and grid-based UWE, as well as the actual precipitation data, all exhibit characteristics of a normal distribution in Fig.5, instead of the typical skewed distribution. This is primarily because this study focuses on summer precipitation over the entire China region, which covers a large area and spans a long period rather than the precipitation in a single grid or station in a short period. Within this range, various types of precipitation events, including light, moderate, and heavy rain, make the probability distribution closer to the normal distribution. Besides, In Fig. 5, probability distribution is calculated based on the monthly anomaly precipitation, and after a treatment of probability function of python program, which smooth the distribution curve. Therefore, the final probability distribution appears as a normal distribution.

**5. The authors investigated the relationship between the dispersal intensity and the ACC, and found that high ACCs correspond to low dispersal intensity. That is, they have negative correlation. From fig.8, most regions over China, such as the northwest region and coastal areas show the negative correlations. However, the negative correlations are not that evident in the middle regions of China. What is the reason, some clarifications are needed.**

**Reply**: Thanks! We have added the relevant discussion in the relevant place in the manuscript.

**The expression in line 489 has been modified as follows:**

In the middle region of China, the negative correlation between the dispersal intensity and TCC is not so evident. This could be due to the limitations of the parameterization schemes in the forecast models for this region, which may result in inaccurate simulations of the diffusion process. Additionally, the diversity of meteorological conditions in the middle region could lead to inconsistencies in the relationship between dispersal intensity and TCC across different areas. For instance, the middle region may be influenced by specific meteorological systems, such as frontal systems and cyclones, which can affect the relationship between dispersal intensity and TCC. Therefore, the positive relationship between the dispersal intensity and the ACC can be found in this study, however this kind of relationship has uncertainty in different areas, which still need to be considered in the operational predictions. These aspects require further detailed investigation.

**Minor comments**

**1. Lines 86-87, the reference format is not correct. Right format is Feng et al. (2013, 2020).**

**Reply**: Thanks! We have standardized the format of references throughout the manuscript.

**2. Lines 170 and 348, before 'etc', there should be a comma.**

**Reply**: Thanks! We have made additions.

**3. Line 202, Fig.2 should be replaced with Fig.1.**

**Reply**: Thanks! We have made the replacement.

**4. For equation (6), what the $xs$, $ys$, $xs$ and $ys$ represent? They should be clarified.**

**Reply**: Thanks! We have made the replacement.

**The expression in line 294 has been modified as follows:**

$\bar{x}_s$ and $\bar{y}_s$ are respectively the average abnormal of observation and prediction

for all the stations.

**5. Line 302 and other parts throughout the manuscript, I don't find what the SYS method represents. Is it the abbreviation? If so, please add its full name at the Line 302. If not, please introduce the SYS briefly in the corresponding section.**

**Reply**: Thanks! We have made additional descriptions.

**The expression in line 250 has been added as follows:**

6) The scheme for systematic error selection (SYS). The arithmetic mean of the model prediction errors over the years is calculated, after which it is superimposed on the model's original prediction results to obtain the systematic error revised prediction of the systematic error revised prediction of the model. This scheme is primarily used for comparison with the other five dynamic-statistic schemes.

**6. Line 428, it is formula (8), not 'formal (5)'.**

**Reply**: Thanks! We have made the replacement.

**7. Line 446, it is quite close, not 'it is quite closet'.**

**Reply**: Thanks! We have made the replacement.